# Effects of an Adapted Sports Intervention on Elderly Women in Need of Long-Term Care: A Pilot Study

**Takashi Kawano** [1,*] , **Goro Moriki** [1] , **Shinya Bono** [1] , **Junya Masumoto** [1] , **Nobuyuki Kaji** [1] , **Hungu Jung** [2]
**and Masahiro Yamasaki** [1]

1   Department of Sports, Health and Well-Being, Faculty of Human Health Science, Hiroshima Bunka Gakuen University, 3-3-20 Heiseigahama, Akigunsakacho, Hiroshima 731-4312, Japan; moriki@hbg.ac.jp (G.M.); bono@hbg.ac.jp (S.B.); masumoto@hbg.ac.jp (J.M.); kaji@hbg.ac.jp (N.K.); myama@hbg.ac.jp (M.Y.)
2   Graduate School of Integrated Arts and Sciences, Hiroshima University, 1-7-1 Kagamiyama, Higashi-Hiroshima, Hiroshima 739-8521, Japan; harazu_21@yahoo.co.jp
*   Correspondence: kawano@hbg.ac.jp; Tel.: +81-82-884-1001; Fax: +81-82-884-0600

**Abstract:** This study aimed to evaluate the effects of an adapted sports intervention on elderly women in need of long-term care (NLTC). Although participation in sports activities positively impacts subjective health status, few studies have evaluated the safety, comfort, and effectiveness of competitive sports in elderly women in NLTC. In this study, ten elderly women in NLTC (age: 80.6 ± 8.2 years) were asked to participate in boccia, a sport adapted to prevent falls. Participants completed the Profile of Mood States 2nd Edition Short-Form and the Medical Outcome Study 36-Item Short-Form Health Survey Version 2. The results showed an improvement in mood states (anger–hostility, tension–anxiety, and total mood disturbance) of elderly women in the NLTC group compared with the control group. Therefore, boccia, an adapted sport, can be considered a safe and competitive option for such women.

**Keywords:** adapted sport; elderly women; need of long-term care

## 1. Introduction

The average life expectancy in Japan for women was 87.45 years old in 2019, and their healthy life expectancy was 75.5 years old, both of which are some of the highest in the world [1]. However, the number of women who select "good/very good" for a subjective assessment of individual health status and self-rated health is low in Japan compared with its internationally reported values [2]. A gap between objective and subjective health is seen and needs resolution.

The effects of sports and physical activity on activities of daily living (ADL) and health-related quality of life (HRQoL) in the elderly have been shown in various previous studies [3–5]. However, studies on elderly people in need of long-term care (NLTC) have reported limited effects of exercise on ADL [6–8], and little concern for the risk of injury and health status deterioration [9]. Consequently, safety and security are prioritized in exercises for the elderly in NLTC, and simple movements such as walking, balance training, and flexibility exercises have been selected [10,11]. A study on exercise habits among the aged population in Japan showed that the number of people who exercised or played sports in the age group between 65 and 74 years has increased, whereas that in the age group of 75 years and above has decreased rapidly [12]. When examining the group of people whose exercise habits are in decline and the reasons behind this deterioration, one cause may be the inability to continue the exercise due to a decline in ADL; this is all the more significant because this age group are enthusiasts of ADL-demanding exercises, such as walking, aerobic dancing, and competitive sports (e.g., table tennis, golf, bowling) [12]. Additionally, a study comparing elderly men and women without exercise habits using self-rated health showed that the HRQoL scores of elderly women were significantly low, especially for

physical health [13]. Considering these facts, there is an urgent need to develop sports programs for elderly women in the NLTC. Therefore, in this study, we focused on adapted sports (AdS).

AdS is derived from the research field of Adapted Physical Activity, that means "a sport in which everyone can participate by devising rules and equipment that take into account the physical abilities of the participants" [14,15]. Previous studies on AdS have demonstrated positive effects on subjective health, such as mood [16] and HRQoL [17,18] in participants with limited physical function. In particular, mood state is an important measure that leads to improvement in homeboundness and participation in activities, such as exercise [19].

Boccia is a sport that is competitive, safe to perform, and even easier to improve in [20]. In addition, it has been shown to have a positive effect on mood states, even in severely impaired individuals [21]. However, to the best of our knowledge, the effect of participation in boccia on the subjective health status of older women in NLTC is unclear.

Therefore, the present study aimed to clarify the effects of AdS on elderly women in NLTC and examine the changes in mood state and HRQoL after intervention with boccia.

## 2. Materials and Methods

This was a pilot study of a non-randomized open label clinical trial. This study was registered with the UMIN Clinical Trials Registry (UMIN-CTR number: UMIN000047091).

### 2.1. Participants

2.1.1. Elderly Women in Need of Long-Term Care Group: NLTC Group

With the cooperation of the local adult day health center in city Y, we recruited our sample of NLTC participants. Thirteen individuals were enrolled in the study. The participants visited the adult health center primarily because of physical disabilities without any verbal communication restrictions. The type of pathology and disability were paralysis due to stroke, osteoarthritis, or fracture. An individual with complete paralysis of the upper limb was not included in the study.

2.1.2. Healthy Elderly Women Group; Healthy Group

For comparison with the NLTC group, we recruited elderly women over the age of 65 years with the cooperation of a community center in city X, all of whom had no physical disabilities and were in good health. Fifteen participants were enrolled in the study.

### 2.2. Exercise Intervention

2.2.1. Adapted Sport Event

We adopted boccia as the AdS in this study, based on its use in Paralympic events. Boccia is an adapted sport established in Europe that can be played indoors by individuals with varying levels of exercise ability and physical disabilities [21]. Previous studies in the elderly have shown that even low-intensity exercise can improve mood and HRQoL [22]. As boccia was originally designed as an indoor sport for people with severe cerebral palsy and severe extremity motor disabilities [20], it is suitable for elderly individuals in need of long-term care. None of the participants in the NLTC and healthy groups had played boccia prior to this study.

2.2.2. Boccia Exercise Intervention

We conducted the boccia exercise intervention once a week over four weeks, which amounted to four times in total. Each participant practiced propelling a ball four times before each exercise session and was subsequently assigned to two teams of three players in order to play in team competitions in two courts. Based on the Japan Boccia Association's competition rules [23], the team competition consisted of six ends and two balls per player at each end. Each participant had 17 throws, including throwing of the jack to initiate each end. To prevent falls, participants in NLTC group were instructed to throw from a seated position

to ensure safety. We set a time limit of 50 min, and all participants finished their turns within the allotted time. The ball used in this study met international competition standards (Japan Boccia Association official ball, Apowa Tech Corp., circumference: $70 \pm 8$ mm, weight: $275 \pm 12$ g).

*2.3. Measurement*

2.3.1. Baseline Variables

To ensure the safety of the participants, their characteristics were assessed before the exercise intervention. The measurement items were age, body mass index, Tokyo Metropolitan Institute of Gerontology Index of Competence (TMIG-IC) [24], Certification of Needed Long-term Care or Support, and bone mineral density tests. The TMIG-IC was used to evaluate instrumental ADL. It is a multidimensional, 13-item scale comprising three subscales: instrumental self-maintenance (five items), intellectual activity (four items), and social role (four items). The response to each item is either "yes" (one point) or "no" (0 point) for a maximum score of 13 points. The necessity of livelihood support increases as the score decreases.

Certification of needed long-term care or support is a unique Japanese scale that determines the extent to which nursing care is required based on physical functions, activities of daily living, and other factors. This scale is classified: "Not certified", "Support level 1", "Support level 2", "Care level 1" (least disabled), "Care level 2", "Care level 3", "Care level 4", and "Care level 5" (most disabled). This scale determines eligibility for long-term care [25].

A bone densitometry system AOS-100SA (Aloka, Tokyo, Japan) was used to measure the bone mineral density of each participant. This system can compute an osteo sono-assessment index (OSI) in the calcaneus and to measure bone mineral density and the t-scores, to how it compares with that is a healthy young adult mean. The t-scores of the participants were classified according to the definition of the World Health Organization (WHO): a t-score of $-1.0$ or above is "normal", a t-score between $-1.0$ and $-2.5$ indicates "osteopenia", and a t-score $-2.5$ or below indicates "osteoporosis" [26].

Household structures and social networks were assessed to determine the social characteristics of the participants. For household structure, we asked for responses from "single-person", "married couple", and "other". We employed the abbreviated version of the Lubben Social Network Scale (LSNS-6) [27] to examine participants' social networks. This scale measures the size of intimate and active networks of family and friends and consists of the following questions: (1) How many relatives do you see or hear from at least once a month? (2) How many relatives do you feel with which you can talk about private matters? (3) How many relatives do you feel close to, such that you can call on them for help? (4) How many friends do you see or hear at least once a month? (5) How many friends do you feel with which you can talk about private matters? (6) How many friends do you feel close to, such that you can call on them for help? The answers were scored from 0 to 5: 0 = none, 1 = one, 2 = two, 3 = three–four, 4 = five–eight, and 5 = nine or more. The maximum score is 30, and a score of 12 or lower indicates that the individual is "at-risk" for social isolation. This scale is widely used internationally to screen for potential social isolations. This assessment was conducted once during the first orientation.

2.3.2. Outcomes

Profile of Moods States

The Profile of Mood States-Short Form 2nd Edition (POMS2-SF) was used to examine the acute changes in participants' mood states [28]. The POMS2-SF is a 35-item questionnaire that assesses mental states using the following seven scales: anger–hostility (AH), confusion–bewilderment (CB), depression–dejection (DD), fatigue–inertia (FI), tension–anxiety (TA), vigor–activity (VA), and friendliness (F). Furthermore, total mood disturbance (TMD), which represents the overall negative mood state, was calculated from the questionnaire. The higher the seven-scale score, the more dominant the mood state indicated.

AH, CB, DD, FI, TA, and TMD are negative moods, whereas VA and F indicate positive mood. We conducted this questionnaire survey twice in total: before (baseline) and after (post) the exercise intervention on the first day.

Health-Related Quality of Life

The Medical Outcomes Study 36-Item Short-Form Health Survey Version 2 (SF-36v2) [29] was administered on the first and last day (one month later) of the intervention period to measure the changes in the participants' health-related quality of life (HRQoL) following the boccia intervention. The SF-36v2 consists of eight subscales to measure health status: physical functioning (PF), physical role functioning (RP), bodily pain (BP), social role functioning (SF), general health perceptions (GH), vitality (VT), emotional role functioning (RE), and mental health (MH). Furthermore, three summary scores can be obtained from the subscales, allowing for further analysis: physical component summary (PCS), mental component summary (MCS), and role/social component summary (RCS). We registered the use of the SF-36v2 with the license holder Qualitest Inc. We conducted this questionnaire survey twice, on the first day (baseline) of the orientation and the last day (Post) of the intervention period.

## 2.4. Ethical Approval

The study protocol was reviewed and approved by the University Research Ethics Committee (Approval No. HS-2018002). All participants were informed of the purpose of this study, its method, as well as the anonymity of the participants, both on paper as well as verbally. It was explained that they were free to end their participation at any given time. Written informed consent was obtained from all participants prior to the study.

## 2.5. Statistical Analysis

The findings of the TMIG-IC, LSNS-6, POMS2-SF, and SF-36v2 were scored. The values are presented as the means and standard deviations with reference to age and bone mineral density measurements. Mann–Whitney U tests were used to compare the TMIG-IC and LSNS-6 scores of the NLTC and healthy groups, and bone mineral density measurements. Chi-square tests were performed to assess between-group differences in the Certification of needed long-term care or support levels. Furthermore, a two-way analysis of variance (two-way ANOVA) was performed to examine the influence of two factors of exercise intervention (baseline and post) and group (NLTC group, healthy group) on the POMS2-SF and SF-36v2 scores of the NLTC and healthy groups. If a significant interaction was found, a post hoc analysis using Bonferroni adjustment was conducted for the exercise intervention factor (baseline, post). Statistical significance was set at a *p*-value of less than 0.05, and SPSS version 24.0 (IBM SPSS, Tokyo, Japan) was used for data analysis.

## 3. Results

The flowchart of the study is shown in Figure 1. Of the thirteen participants enrolled in the NLTC group, one declined to participate and two discontinued participation during the intervention period. Of the fifteen participants in the healthy group, one withdrew, and three discontinued participation during the intervention period. One healthy participant who had missing questionnaires was excluded from the analysis. After the 4-week study period, the data from the remaining 20 participants were analysed.

## 3.1. Baseline Characteristics

Table 1 shows the baseline characteristics of the participants. The TMGI-IC score of the NLTC group ($9.2 \pm 3.7$) was significantly lower than that of the healthy group ($12.6 \pm 0.7$) ($p < 0.05$). A significant difference in the certification of needed long-term care or support levels was observed between the groups. Specifically, the scores of the NLTC group were low in social role ($p < 0.05$). Furthermore, the results of bone mineral density testing showed no osteoporosis, but seven out of the ten participants in the NLTC group

and six out of the ten participants in the healthy group fell within the range indicating osteopenia. Additionally, the household composition in the control group were similar, but the LSNS-6 score was significantly lower in the NLTC group ($p < 0.001$). The mean LSNS-6 score for the NLTC group was $11.3 \pm 5.4$, with four participants below the cutoff score for social isolation.

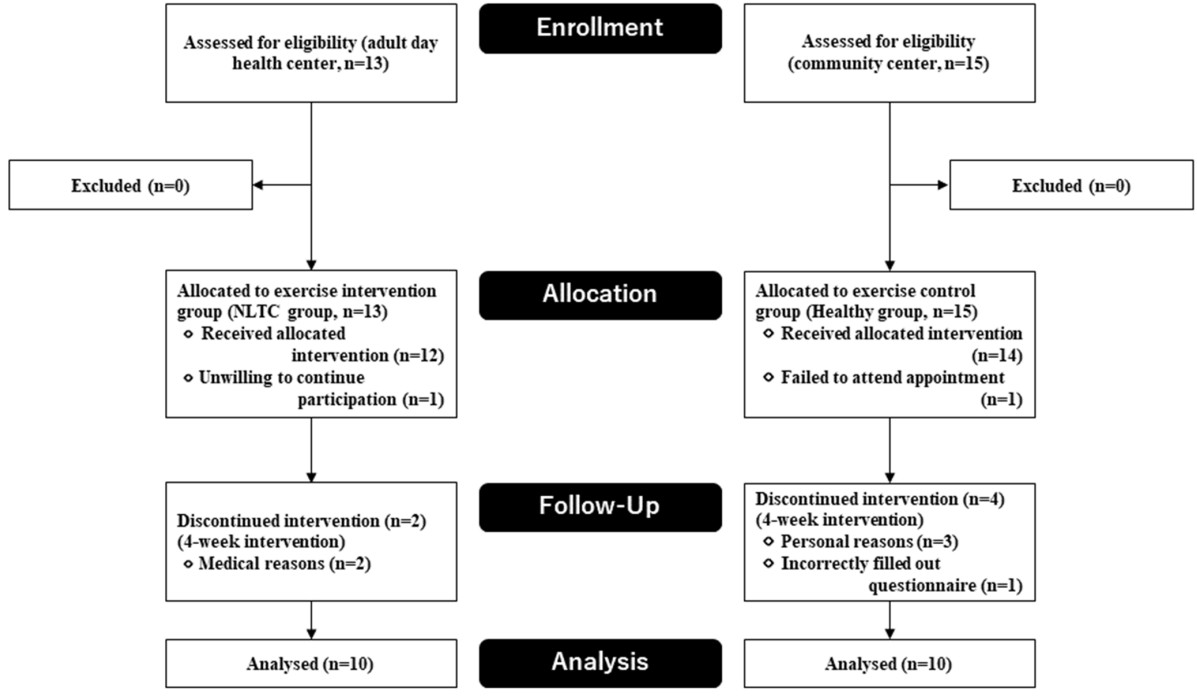

**Figure 1.** Flow diagram of the participants.

**Table 1.** Baseline characteristics of the participants.

| Variables | NLTC Group (*n* = 10) | | Healthy Group (*n* = 10) | | *p* Value |
|---|---|---|---|---|---|
| Age (years) | 80.6 | ±8.2 | 81.8 | ±5.2 | |
| Body mass index (kg/m²) | 22.5 | ±2.2 | 22.2 | ±3.4 | |
| Total score of TMIG-IC (range, 0–13) | 9.2 | ±3.7 | 12.6 | ±0.7 | * a |
| Instrumental Self-Maintenance (range, 0–5) | 3.5 | ±1.9 | 5.0 | ±0.0 | |
| Intellectual Activity (range, 0–4) | 3.4 | ±0.8 | 3.9 | ±0.3 | |
| Social Role (range, 0–4) | 2.3 | ±1.4 | 3.7 | ±0.7 | * a |
| Certification of needed long-term care or support | | | | | ** b |
| Not certified | 0 | 0% | 10 | 100% | |
| Support levels 1 | 4 | 40% | 0 | 0% | |
| Support levels 2 | 2 | 20% | 0 | 0% | |
| Care levels 1 | 2 | 20% | 0 | 0% | |
| Care levels 2 | 2 | 20% | 0 | 0% | |
| Care levels 3 | 0 | 0% | 0 | 0% | |
| Care levels 4 | 0 | 0% | 0 | 0% | |
| Care levels 5 | 0 | 0% | 0 | 0% | |
| Quantitative ultrasound | | | | | |
| Speed of sound (m/s) | 1491.6 | ±13.4 | 1505.4 | ±10.7 | * a |
| Broadband ultrasound attenuation (-dB/MHz) | 69.3 | ±6.3 | 67.5 | ±10.2 | |
| T-score (SD) | −1.1 | ±0.5 | −1.0 | ±0.6 | |
| Normal (T-score more than −1.0) | 3 | 30% | 4 | 40% | |
| Osteopenia (T-score between −2.5 and −1.0) | 7 | 70% | 6 | 60% | |
| Osteoporosis (T-score less than −2.5) | 0 | 0% | 0 | 0% | |
| Household composition | | | | | |
| Single person | 2 | 20% | 2 | 20% | |
| Married couple | 2 | 20% | 3 | 30% | |
| Other | 6 | 60% | 5 | 50% | |
| Lubben Social Network Scale-6 score (range, 0–30) | 11.3 | ±5.4 | 20.2 | ±3.5 | *** a |
| Normal (score more than 12) | 6 | 60% | 10 | 100% | |
| Social isolation (score less than 12) | 4 | 40% | 0 | 0% | |

Data are expressed as Mean ± SD values, Number, %, *: $p < 0.05$, **: $p < 0.01$, ***: $p < 0.001$, a: Mann–Whitney $U$ test, b: Chi-square test.

### 3.2. Outcomes

Table 2 shows POMS-SF and SF-36v2 scores at baseline and at 4 weeks. The POMS2 score (AH, FI, TA, F, TMD) presented significant interaction effects. The NLTC group expressed significant improvements in AH ($p < 0.001$), TA ($p < 0.05$), TMD ($p < 0.01$). Among them, TMD showed a remarkable positive change of 15.3 points. In contrast, the healthy group showed significant worsening in FI ($p < 0.05$) and F ($p < 0.05$).

**Table 2.** Outcome of the exercise intervention.

| Variables | NLTC Group (n = 10) Baseline | Post | | Healthy Group (n = 10) Baseline | Post | | Interaction (Group × Time) F | p Value | | η² |
|---|---|---|---|---|---|---|---|---|---|---|
| | \multicolumn Score of POMS2-SF (Profile of Mood States) | | | | | | | | | |
| AH | 54.4 ±2.6 | 45.4 ±1.5 | *** | 53.2 ±4.3 | 53.2 ±4.3 | | 9.643 | 0.006 | ** | 0.349 |
| CB | 63.2 ±6.3 | 56.8 ±5.1 | | 56.4 ±4.7 | 56.4 ±4.7 | | 1.075 | 0.314 | | 0.560 |
| DD | 65.8 ±6.4 | 49.9 ±3.2 | | 62.7 ±5.8 | 51.1 ±4.2 | | 0.527 | 0.477 | | 0.028 |
| FI | 55.7 ±5.3 | 45.1 ±2.1 | | 49.9 ±4.0 | 63.3 ±2.8 | * | 7.825 | 0.012 | * | 0.303 |
| TA | 58.8 ±5.7 | 43.6 ±2.3 | * | 51.1 ±4.2 | 62.0 ±4.0 | | 8.035 | 0.011 | * | 0.309 |
| VA | 48.1 ±3.0 | 61.7 ±1.9 | | 63.3 ±2.8 | 62.7 ±5.8 | | 2.428 | 0.137 | | 0.119 |
| F | 50.4 ±2.9 | 58.3 ±2.3 | | 62.0 ±4.0 | 49.9 ±4.0 | * | 7.209 | 0.015 | * | 0.286 |
| TMD | 59.5 ±5.5 | 44.2 ±2.3 | ** | 49.6 ±4.8 | 49.6 ±4.8 | | 7.611 | 0.013 | * | 0.297 |
| | \multicolumn Score of SF-36v2 (HRQOL) | | | | | | | | | |
| PF | 19.6 ±7.5 | 20.3 ±6.2 | | 44.5 ±4.4 | 44.8 ±3.8 | | 0.007 | 0.934 | | 0.000 |
| RP | 34.1 ±5.1 | 35.1 ±4.3 | | 40.1 ±3.8 | 46.4 ±3.5 | | 0.485 | 0.495 | | 0.026 |
| BP | 40.5 ±3.4 | 40.1 ±3.3 | | 43.2 ±3.3 | 49.7 ±3.1 | * | 4.710 | 0.044 | * | 0.207 |
| GH | 44.7 ±4.2 | 52.0 ±3.9 | | 51.4 ±2.9 | 56.4 ±2.8 | | 0.520 | 0.480 | | 0.028 |
| VT | 45.7 ±5.0 | 47.6 ±2.9 | | 55.0 ±2.9 | 56.9 ±2.8 | | 0.000 | 1.000 | | 0.000 |
| SF | 41.5 ±5.4 | 48.0 ±3.4 | | 50.6 ±2.9 | 49.9 ±3.1 | | 2.333 | 0.144 | | 0.115 |
| RE | 40.3 ±3.6 | 41.9 ±4.9 | | 43.6 ±4.2 | 49.8 ±3.1 | | 0.758 | 0.395 | | 0.040 |
| MH | 41.9 ±5.5 | 50.8 ±3.7 | | 53.7 ±2.3 | 56.4 ±2.6 | | 2.088 | 0.166 | | 0.104 |
| PCS | 27.3 ±5.2 | 25.3 ±5.1 | | 41.1 ±4.5 | 45.3 ±3.0 | | 1.986 | 0.176 | | 0.099 |
| MCS | 52.7 ±4.5 | 59.7 ±2.7 | | 58.3 ±1.8 | 60.2 ±2.5 | | 1.576 | 0.225 | | 0.080 |
| RCS | 41.5 ±3.7 | 43.7 ±4.1 | | 43.7 ±4.5 | 46.2 ±3.1 | | 0.002 | 0.962 | | 0.000 |

Data are expressed as Mean ± SD values, *: $p < 0.05$, **: $p < 0.01$, ***: $p < 0.001$, AH: Anger–Hostility, CB: Confusion–Bewilderment, DD: Depression–Dejection, FI: Fatigue–Inertia, TA: Tension–Anxiety, VA: Vigor–Activity, F: Friendliness, TMD: Total Mood Disturbance, PF: Physical functioning, RP: Role physical, BP: Bodily pain, GH: General health, VT: Vitality, SF: Social functioning, RE: Role emotional, MH: Mental health, PCS: Physical component summary, MCS: Mental component summary, RCS: Role/Social component summary.

In the SF-36v2 score, only BP ($p < 0.05$) showed a significant interaction effect, whereas the healthy group showed significant worsening in BP ($p < 0.05$).

## 4. Discussion

This study aimed to evaluate the effects of AdS boccia on mood state and HRQoL in elderly women requiring long-term care. The baseline of the targeted NLTC group was significantly lower than that of the healthy group, not only in physical functioning, but also in social connectedness. The results of the boccia intervention for the NLTC group showed significant improvements in negative mood.

In Japan, the number of people who engage in sports activities to maintain health is increasing annually. Nevertheless, it has been pointed out that the perception of aging and decline in physical function is associated with retirement over sports activity [30], and our previous study also demonstrated the relationship between subjective health and sports activity [12]. Therefore, it is important to provide exercise that can be performed safely even in a state of needing care, as well as to devise exercise content to improve subjective health, such as mood state and HRQoL. This study is the first to examine the effects of AdS, such as boccia, on the subjective health status of elderly women in NLTC compared with healthy elderly women of the same age range.

Interestingly, this study showed significantly and positively increased mood states in elderly women with NLTC compared with healthy elderly women. This is a novel finding of this study. In a previous study on gender and subjective health in elderly Japanese women, positive social integration and social networks were associated with better subjective health and no risk of falling [31,32]. The improvement in mood state status in this study indicates that, in elderly women, even those who require care, boccia can be a useful tool to increase subjective health status.

This study has some limitations. First, the sample size is small. Despite this, statistically significant differences, which could be exhibited in a larger sample size, were found. Therefore, our study may have clinical significance for geriatric patients. Second, this study was a very simple examination. Each of the ten NLTC participants in this study had unique needs, but this pre-assessment was limited to physical items. When implemented as an adapted physical activity for people with disabilities, evidence-based and appropriately designed and modified equipment, task criteria, instructions, physical and social environments, and rules must be provided [33]. Implementing and assessing only quantitative effects, without a detailed consideration of individual needs runs the risk of reducing options and possibilities in practice settings that are susceptible to heuristic influences [34]. In this study, although we used only a kind of sports activity, such as boccia, with a short observation time, it was demonstrated that it can be conducted at a local day service center or community center. A future study will be based on a more detailed evaluation to assess the impact of AdS, including various types of sports activity, on NLTC in a long-term observation with follow-up.

## 5. Conclusions

Elderly women who require care can choose and enjoy competitive sports. Despite its low exercise intensity and small sample size, boccia, an adapted sport that eliminates risk factors for falls, may improve subjective health conditions, such as mood state, in the short term. A further study with a large sample will be needed for a more extended period.

**Author Contributions:** Conceptualization, T.K. and N.K.; methodology, G.M., S.B. and M.Y.; validation, J.M. and H.J.; formal analysis, T.K.; investigation, T.K. and N.K.; data curation, T.K. and G.M.; writing—original draft preparation, T.K.; writing—review and editing, J.M.; visualization, H.J.; supervision, M.Y.; project administration, S.B.; funding acquisition, T.K. and M.Y. All authors have read and agreed to the published version of the manuscript.

**Funding:** This work was supported by JSPS Grant-in-Aid for Scientific Research(C) Grant Number 21K02065.

**Institutional Review Board Statement:** The study was conducted according to the guidelines of the Declaration of Helsinki, and approved by the Department Research Ethics Committee of Hiroshima Bunka Gakuen University (protocol code HS-2018002).

**Informed Consent Statement:** Informed consent was obtained from all subjects involved in the study.

**Data Availability Statement:** All data are presented in the manuscript.

**Acknowledgments:** We are grateful to the participants in this study. We also thank Masako Kuramoto and Mika Kuramoto for their expert technical assistance in writing the manuscript.

**Conflicts of Interest:** The authors declare no conflict of interest.

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
