# Peer review of "Effects of an Adapted Sports Intervention on Elderly Women in Need of Long-Term Care: A Pilot Study"

_applsci, doi:10.3390/app12063097_

Round 1

Reviewer 1 Report

The topic is original and potentially useful in clinical practice. However, significant revisions are required and the manuscript is not suitable for publication in its present format.

First of all the study design is wrong: this is not a case report, but a non-randomized open label clinical trial.

The Introduction does not contain exhaustive background justifying the study’s hypotheses: the Authors generally talk about “positive effects” of AdS, without specifying what they mean (physical parameters? Cardiovascular performance? Quality of life? Etc.). Thus, the study’s aim is not clear. This confusing presentation is reflected in the Methods, where is very difficult to understand what are the outcome measures and the timing of the assessments.

Moreover, inclusion and exclusion criteria are missing; it is not clear the heterogeneity in number of the eligible participants in the two groups (15 vs 13); the definition of “women in need of long-term care” doesn’t provide adequate information about the type of pathology and disability.

The Authors consider many variables (physical, social, mood, QoL, cardiovascular), with too many assessment measures (age, BMI, TMIG-IC, BMD, LSNS-6, POMS2-SF, etc.) and this is very confusing for the reader, since it is not clear what are the baseline characteristics and what are the outcome measures.

Author Response

Responses to Reviewer 1

We thank reviewer for careful reading my manuscript and for your helpful suggestions. Our responses to the referees’ comments are as follow:

Suggestion (1)

The study design is wrong: this is not a case report, but a non-randomized open label clinical trial.

[Response] As you pointed out, this study is not a case report; we will change it to "Brief report" (p.1, line 1). We have also added a note that This study was a non-randomized open label clinical trial (p.2, line 67), and furthermore, we have specified how the participants were enrolled (p.8, figure 1).

Suggestion (2)

The Introduction does not contain exhaustive background justifying the study’s hypotheses: the Authors generally talk about “positive effects” of AdS, without specifying what they mean (physical parameters? Cardiovascular performance? Quality of life? Etc.). Thus, the study’s aim is not clear. This confusing presentation is reflected in the Methods, where is very difficult to understand what are the outcome measures and the timing of the assessments.

[Response] We have written a detailed report on previous research on adapted sports (p.2, line 52-62). In particular, we have added information about the effects on mood states and health-related quality of life that this study addresses. Furthermore, we clearly stated the purpose of the study in the Abstract (p.1, line 14-15) and Introduction (p.2, line 63-64). Methods were limited to variables relevant to the results (p.3, line 106-173).

Suggestion (3)

Moreover, inclusion and exclusion criteria are missing; it is not clear the heterogeneity in number of the eligible participants in the two groups (15 vs 13); the definition of “women in need of long-term care” doesn’t provide adequate information about the type of pathology and disability.

[Response] For the sake of clarity, these two groups were changed to the group of elderly women in need of care (NLTC group) and the group of healthy elderly women (Healthy group) instead of the A/B group (p.2, line 70 and 77). In addition, we have also added the pathogenesis and types of diseases of NLTC group members (p.2, line 74-75).

Suggestion (4)

The Authors consider many variables (physical, social, mood, QoL, cardiovascular), with too many assessment measures (age, BMI, TMIG-IC, BMD, LSNS-6, POMS2-SF, etc.) and this is very confusing for the reader, since it is not clear what are the baseline characteristics and what are the outcome measures.

[Response] As you pointed out, the variables measured in this study were divided into baseline (p.3, line 107-144) and outcome (p.3, line 146-173), and variable information not relevant to the discussion, such as HRV, was removed. Instead, I explained the exercise intensity of boccia based on previous studies (p.2, line 86-88).

Reviewer 2 Report

This manuscript titled: Effect of Adapted Sports on the self-rated Health of elderly women in need of long-term care gives not answer on important aspect of this study - the effects of AdS exercise intervention on elderly women in need of long-term care. 

1/ Authors did not interpret properly adopted physical activity. this concept because adaptation to physical activity opportunities is most often provided in form of appropriately designed and modified equipment, task criteria (e.g., modifying skill quality criteria), instructions (e.g., using supports, peer tutors, non-verbal instructions, motivational strategies), physical and social environments (e.g., increasing or decreasing court dimensions; segregated vs. inclusive: mastery oriented, collaborative or competitive social environment; degree of peer support). In my opinion authors approached the adopted physical activity too briefly and easily (they chose only one discipline, poorly motivating this choice). Additionally,  incorrectly they described the methodology of this physical activity.
2 / Very small research group and poorly described
3 / On this basis, the results are not reliable and not significant. It is rather an attempt at the feasibility of the study.

Author Response

Responses to Reviewer 2

We thank reviewer for careful reading my manuscript and for your helpful suggestions. Our responses to the referees’ comments are as follow:

Suggestion (1)

This manuscript titled: Effect of Adapted Sports on the self-rated Health of elderly women in need of long-term care gives not answer on important aspect of this study - the effects of AdS exercise intervention on elderly women in need of long-term care.

[Response] In response to your suggestion, we have changed our title to “Effects of an adapted sports intervention on elderly women in need of long-term care” (p.1, line 2-3).

Suggestion (2)

Authors did not interpret properly adopted physical activity. this concept because adaptation to physical activity opportunities is most often provided in form of appropriately designed and modified equipment, task criteria (e.g., modifying skill quality criteria), instructions (e.g., using supports, peer tutors, non-verbal instructions, motivational strategies), physical and social environments (e.g., increasing or decreasing court dimensions; segregated vs. inclusive: mastery oriented, collaborative or competitive social environment; degree of peer support). In my opinion authors approached the adopted physical activity too briefly and easily (they chose only one discipline, poorly motivating this choice). Additionally, incorrectly they described the methodology of this physical activity.

[Response] Thank you very much for your valuable comments. We have looked at the previous studies you mentioned. As you pointed out, we think that some nursing homes in Japan are providing unsubstantiated support by influencing heuristics without fully evaluating the alternatives. Originally, the purpose of this study was to refute the common belief that competitive sports are difficult for elderly people who need care.

In the Introduction, we have added information about adapted physical activities and adapted sports (p.2, line 52-62). In particular, we added an explanation of why we chose boccia and the variables to be analyzed (p.2, line 54-58). Therefore, the description was modified to more clearly indicate the purpose and methods of this study. Furthermore, we added the pathology of the subjects, the type of disability, and the flowchart of the experiment (p.8, figure 1). In addition, the description of indications in APA was missing, so we added it to the discussion (p.6, line 253-259).

Suggestion (3)

Very small research group and poorly described

[Response] In response to your suggestion, we re-examined the results with non-parametric tests (p.4, line 183-195).

Suggestion (4)

On this basis, the results are not reliable and not significant. It is rather an attempt at the feasibility of the study.

[Response] Although the sample size was small, the results showed a significant difference. We believe that we have obtained certain results. However, since this is a brief report as a study, we added the words "the sample size is small. Despite this, statistically significant differences, which could be exhibited in a larger sample size, were found" as a limitation (p.6, line 249-251).

Reviewer 3 Report

Overall, I think this paper is a well-written study. Therefore, my opinion is that it may be published after a minor correction.

1. A text to further emphasize the purpose of this study should be added.
2. A specific description of why the application of AdS is important and necessary.
3. Boccia's description of exercise intensity is missing.
4. There is a typo where such as is written twice in Conclusion.

Author Response

Responses to Reviewer 3

We thank reviewer for careful reading my manuscript and for your helpful suggestions. Our responses to the referees’ comments are as follow:

Suggestion (1)

A text to further emphasize the purpose of this study should be added.

[Response] I clearly stated the purpose of the study in the Abstract (p.1, line 14-15) and Introduction (p.2, line 63-65). Methods were limited to variables relevant to the results. (p.3, line 107-144, 146-173)

Suggestion (2)

A specific description of why the application of AdS is important and necessary.

[Response] We believe that not only exercise, but also competitive sports are important for elderly people who need care. We have added a lot of information in the introduction to show our hypothesis that adapted sports are effective for the health of elderly people who need care (p.2, line 52-62).

Suggestion (3)

Boccia's description of exercise intensity is missing.

[Response] Since HRV does not affect the discussion of this study, the variable information was removed, and previous studies were used to explain the exercise intensity (p.2, line 86-88).

Suggestion (4)

There is a typo where such as is written twice in Conclusion.

[Response] We are so embarrassed to have made such a rudimentary mistake. We carefully checked the whole thing (p.6, line 266).

Round 2

Reviewer 1 Report

The Authors revised the text and the overall readibility and comprehension in now improved. However some revisions are still necessary before publication.

  • Page 2, line 58: "Therefore, we adopted boccia as the AdS in this study". I suggest to move this sentence from the Introduction to the Methods.
  • Page 2, Methods: inclusion and exclusion criteria are still missing. The Authours should specify elegibility criteria for the study: for instance, I supposed that a patient with a complete paralysis of upper limb could not be included in the study.
  • Page 2, line 72: "Thirteen individuals were enrolled in the study". Line 80 "Fifteen participants were enrolled in the study". Please, move these sentences to the Results.
  • Results section, Outcomes (lines 219-224): please give a comprehensive explanation of the results reported in table 2.  For instance, no mention about the comparison with healthy group has been made.
  • Conclusion: based on the study's data, the small sample size and the lack of a medium-term and long-term follow-up, the assertion "boccia [...] improves subjective health conditions, such as mood state" can not be proved.

Author Response

Responses to Reviewer 1

We wish to express our strong appreciation to the reviewers for their insightful comments on our manuscript. We feel the comments have helped us significantly improve the paper. Our responses to the reviewers' comments are as follow:

Suggestion (1)

Page 2, line 58: "Therefore, we adopted boccia as the AdS in this study". I suggest to move this sentence from the Introduction to the Methods.

[Response] In response to your suggestion, we have moved the explanation regarding boccia to the Methods section (p.2, line 85).

Suggestion (2)

Page 2, Methods: inclusion and exclusion criteria are still missing. The Authours should specify elegibility criteria for the study: for instance, I supposed that a patient with a complete paralysis of upper limb could not be included in the study.

[Response] As you pointed out, the inclusion/exclusion criteria remained unclear. Therefore, "An individual with complete paralysis of the upper limb was not included in the study" was added in the relevant section. (p.2, line 75-76)

Suggestion (3)

Page 2, line 72: "Thirteen individuals were enrolled in the study". Line 80 "Fifteen participants were enrolled in the study". Please, move these sentences to the Results.

[Response] In response to your suggestion, we have moved it to the Results (p.5, line 197, 199).

Suggestion (4)

Results section, Outcomes (lines 219-224): please give a comprehensive explanation of the results reported in table 2.  For instance, no mention about the comparison with healthy group has been made.

[Response] As you indicated, we have added the Results of the analysis for the health group (p.5, line 221-224).

Suggestion (5)

Conclusion: based on the study's data, the small sample size and the lack of a medium-term and long-term follow-up, the assertion "boccia [...] improves subjective health conditions, such as mood state" can not be proved.

[Response] We agree that this point requires clarification, and have modified the Conclusion (p. 6, lines 266-268).

Reviewer 2 Report

1/In my opinion this is only a pilot study. Authors should include this information in title and during manuscript. 

2/ This study was conducted using only one sport: boccia. Therefore authors should not use term: adopted sports but only boccia.

3/ Short time of observation.

Author Response

Responses to Reviewer 2

We wish to express our strong appreciation to the reviewers for their insightful comments on our manuscript. We feel the comments have helped us significantly improve the paper. Our responses to the reviewers' comments are as follow:

Suggestion (1)

In my opinion this is only a pilot study. Authors should include this information in title and during manuscript. 

[Response] In response to your suggestion, we have added "A pilot study" to the subtitle (p.1, line 3) and Methods (p.2, line 66).

Suggestion (2)

This study was conducted using only one sport: boccia. Therefore authors should not use term: adopted sports but only boccia.

Suggestion (3)

Short time of observation.

[Response] As you pointed out, this study was conducted using boccia only and is a short-term intervention. Therefore, we have reflected this in the Discussion (p. 6, line 258-262) and modified the Conclusions (p.6, line 266-268).

Round 3

Reviewer 1 Report

The Authors modified the paper based on the suggestions.

In my opinion the manuscript is publishable in the present form.

Reviewer 2 Report

I accept in current form.